# Non-Fatal Occupational Injury Prevalence and Associated Factors in an Integrated Large-Scale Textile Industry in Addis Ababa, Ethiopia

**DOI:** 10.3390/ijerph19063688

**Published:** 2022-03-20

**Authors:** Hailemichael Mulugeta, Abyneh Birile, Hilina Ketema, Muluken Tessema, Steven M. Thygerson

**Affiliations:** 1Department of Public Health, College of Health Science, Debre Berhan University, Debre Berhan P.O. Box 445, Ethiopia; abaybrl@gmail.com (A.B.); hilinaketema13@gmail.com (H.K.); mklitmuluken@gmail.com (M.T.); 2Department of Public Health, College of Life Sciences, Brigham Young University, Provo, UT 84602, USA

**Keywords:** non-occupational injury, associated factors, integrated textile industry, Ethiopia

## Abstract

Occupational injuries disproportionately impact workers of the textile industry in low-income countries. The present study investigates the prevalence of non-occupational injury and its associated factors among workers in an integrated textile industry in Addis Ababa, Ethiopia. A cross-sectional study was conducted from 17–26 May 2021. A total of 311 workers were eligible for participation. The information was collected through an interview-administered questionnaire. The findings were presented with descriptive statistics and the relationship among variables was assessed with multi-variable analyses. A total of 291 (93.6%) participants were interviewed. The prevalence of non-fatal occupational injury was 11% [95% CI: 7.7–15.5] in the past 12 months. The hands and fingers were the most affected body parts. Male gender [AOR: 3.40; 95% CI (1.13–10.5)], the age group of 18–29 years [AOR: 6.69; 95% CI (1.35–32.7)], sleeping less than seven hours in a night [AOR: 2.67; 95% CI (1.03–6.97)], machine-based jobs [AOR: 3.59; 95% CI (1.02–12.6)], the workplace housekeeping [AOR: 5.87; 95% CI (1.45–23.8)], and inadequate empowerment to prevent injury accident [AOR: 4.6; 95% CI (1.01–20.9)] were associated factors with occupational injury. The prevalence of non-fatal occupational injuries is lower than the previous studies among textile workers. As a result, improving workplace safety, changing sleeping habits, and empowering workers to participate in injury prevention should be a priority in intervention.

## 1. Introduction

Textile industries are highly expanded in low-income countries including sub-Saharan Africa [1]. The integrated textile factories comprise both textile production and garment processing [2].

There are numerous dangers and risks for workers, ranging from noise and hazardous substances to manual handling and working with potentially dangerous machinery in textile industries. These industries are the most common manufacturing businesses with a high prevalence of work-related injuries [3,4,5,6]. Occupational injuries disproportionately impact young and productive age groups, resulting in significant economic losses for victims, their families, and nations as a whole [7,8]. Recent studies reported a one-year prevalence of 28.3% in Bangladesh [4], 31.4–42.7% in Ethiopia [9,10], and 65.8% in Turkey [11]. Cut, fracture, puncture, abrasion/laceration, and eye injury are the most reported types of injury among the sectors [9,10,11,12]

Occupational health and safety (OHS) vulnerability studies reported that workers with inadequate workplace OHS policies and procedures, awareness of rights and responsibilities, and discouragement of worker participation in safety to prevent injury were predisposing factors resulting in workplace injury [13,14]. Young workers (<30 years), male gender, working >48 h/week, handling objects >20 kg, visual concentration needed for the task, poor maintenance of machine, health and safety training, job stress, and sleep disorders among textile workers are significant predictors of occupational injury [5,15]. Similarly, the risk of occupational injuries are also related to carelessness, rushing, and improper physical condition in the work place (Floor and untidiness) and are causes of work place injury [11]. Most of those factors are related with safety practices [16].

Ethiopia’s growth and transformation Plan II and Ethiopia’s 2015 National Development Strategy single out the textile and leather industries as priority manufacturing industries [17]. The country is attracting the attention of companies due to the availability of cotton, cheap workforce, and low-cost energy supply [18]. More than 200 garment and textile industries are currently functional and have an average growth rate of 51% from 2013 to 2018. Textile and garment products represent 17% of total manufacturing value, and the sector is expected to grow by around 40% per year in the next few years. The estimated workers in these sectors number over 62,000 and more than 60% are female [19].

Though Ethiopia’s textile industry is rapidly growing, related workplace injury data are scarce. This study aimed to assess the prevalence of non-fatal occupational injury and its associated factors among workers in an integrated textile industry in Addis Ababa, Ethiopia.

## 2. Materials and Methods

### 2.1. Study Setting and Period

An institution-based cross-sectional study was conducted from 17 to 26 May 2021 among integrated textile industry workers in Addis Ababa, the capital of Ethiopia. The industry has more than 50 years of experience in the knitting, blanket, garment, and dying process. More than 800 workers were engaged in the administrative, marketing, and production process during the study period.

### 2.2. The Study Population

In this study, all workers greater than 18 years old, with direct involvement in the production process, and with more than one year of work experience in the same production activities were eligible. Excluded workers were those who were not able to verbally communicate, managers, and/or heads of department. A total of 311 workers (70 blanket, 21 knitting, 189 garment, 22 dying and 9 utility or supporters of the production) were involved in the study.

### 2.3. Study Variables

Injury status was the dependent variable. Independent variables were socio-demographic characteristics: sex, age, education, marital status, employment pattern, work experience, and salary [5,9,10,15]; personal behavior: sleeping hour/day, alcohol use, smoking, personal protective equipment utilization, and khat chewing [15,20]; psychosocial factors: workplace stress, and job satisfaction [20]; and work environment: working hours per week, job category, work area (light, space and neatness), machine safeguarding, exposure status to hazard, OHS policies and procedures; OHS awareness and OHS empowerment [5,9,10,11,20,21].

### 2.4. Data Collection Tool and Procedure

The data collection tools included a questionnaire developed from previously published papers in a peer reviewed journal [5,11,14,21,22,23,24,25,26]. The questionnaire focused on sociodemographic, behavioral, psychosocial, and environmental variables. As part of the questionnaire, worker stress was assessed using the Marlin Company and the American Institute of Stress scale [27]. The American Institute of Stress validated the tool, which is now utilized in a variety of occupations [20,28,29,30,31]. A 27-item validated OHS vulnerability questionnaire was used to assess hazard exposure, OHS policies and procedures, awareness, and empowerment [21,22,26]. Hazard exposure refers the workers’ exposure to the materials and task-related hazard. Measurements included questions about how often the worker is exposed to a variety of risk factors on a scale of never to every day. OHS policies and procedures are defined as regular training, communication, systems to identify hazard, emergency response, and evaluating the workers’ understanding of OSH procedures. This variable was measured on a scale of strongly agree to strongly disagree. OHS awareness refers to individuals’ awareness of safety hazards how to perform their job safely and was measured on a scale of strongly agree to strongly disagree. OHS empowerment refers to the extent to which individuals feel free to voice health and safety concerns, ask questions about health and safety, and refuse unsafe duties and was measured on a scale of strongly agree to strongly disagree [14,22,26,32]. The questionnaire was entered to KoBoToolbox and deployed to collect the data using tablets. The questionnaires were administered through face-to-face interviews with individual workers. The study was conducted after having ethical clearance from Debre Berhan University College of Health Sciences Institutional Review Board. Before performing data collection, permission was granted by industry owners and managers, verbal consent was obtained from each study participant for the study and dissemination of the results. Participation was fully voluntary and confidentiality was ensured for the collected information through the use of coding.

### 2.5. Data Quality Control

Four experienced professionals in occupational health participated in the data collection. Two days of training were given about the study objective, KoBoToolbox software use, data items, interview approach, and ethical issues. The validity of the questionnaire was evaluated by consultation with occupational health experts and pretested before data collection to understand the practical situation before the actual data collection. Internal consistency of the tool was ascertained by pretesting among 16 garment industry workers in another industry, Addis Ababa. Lessons learned during pretesting were used to improve the clarity of the questionnaire and interview approach. Cronbach’s alpha was calculated for the 27-item OHS vulnerability questionnaire and found to be sufficient (α = 0.76).

Data collection activities were conducted in isolated areas (such as offices and separate places from the work station) to maintain the privacy of the respondent and to avoid disturbance [20]. The data collection process was supervised by the principal investigator and the daily collected data were sent to the computer’s server to check for completeness and consistence by the research team.

### 2.6. Data Analysis

The data were exported to Statistical Package for the Social Sciences version 20 (IBM, Armonk, NY, USA) for cleaning and analysis. Bivariate analysis was used to select the candidate variables for multivariable logistic regression model to avoid the cofounder variables [33,34,35,36]. Hence, independent variables with a *p*-value below 0.3 with the outcome variable in the bivariate analysis were included for the multivariable logistic regression model. The required assumptions of the logistic regression were checked with Hosmer and Lemeshow fit test statistics (chi-squared test, *X*^2^ = 3.82, with a degree of freedom of 8 and a significance equal to 0.93). A multi co-linearity assumption was checked. The result revealed a variance inflation factor <1.3 and tolerances >0.8. Odds ratios with 95% confidence interval (CI) were used to declare the presence and the strength of association in the multivariable analysis. Variables with a *p*-value of less than 0.05 were considered statistically significant in the final model.

### 2.7. Operational Definitions and Measurement

The context of variables might vary from study to study. The following terms were operationally specified for the study’s plausibility:

Injury: any physical injury condition sustained by the worker in connection with the performance of their work and received injury care at the clinic level in last 12 months [5].

Workplace stress status: A result with a total score of 26 or more for work stress presence and less than 26 for no work stress [20,24].

Job satisfaction status: A self-report of participants regarding their feelings about their job and if it is pleasurable for them [37].

Khat chewing: The practice of chewing khat leaves by the worker at least once per week for different purposes [15]. Khat is classified as an addictive substance which stimulates and increases excitement that later can affect the central nervous system [38].

Smoking: Referred to smoking at least one stick of tobacco cigarette each day [20].

Alcohol drinking: Referred to consumption of any alcohol for more than twice a week [20].

Adequate work space: Is the working area with enough clear space (walkways) to allow physical actions needed to perform the task [20].

Obstacle-free floor: Includes floors maintained to be free of slip and trip hazards (trailing cables, uneven edges, or broken surfaces) and walkways clear and free from tools and materials [20].

Adequate light: Includes sufficient lighting to allow workers to see machinery movements, controls, displays, and to move about easily and free of flicker and glare [20].

Exposed to hazards: If they reported weekly or more frequent exposure to any two hazards related to the questions of the 11 items [21,22].

Inadequate protections: If workers reported disagree or strongly disagree with at least one of the related statements (policies and procedures, 7 items, awareness, 6 items, or empowerment, 5) [13,21,22].

Personal Protective Equipment (PPE)**:** Includes goggles, helmet, face shield, gloves, boots, and specialized clothing that is designed to protect parts of the body including the eyes, face, hands, body, and feet [5,15].

## 3. Results

### 3.1. Socio-Demographic Characteristics of Study Participants

A total of 291 (93.6%) and 10 eligible workers did not agree to participate during data collection. Of the 291 participants, 26.5% male and 73.5% female participants were interviewed. The participants’ mean age was 32.7 (SD, ±12.1) years, 113 (38.8%) having worked for a duration of less than 3 years and 271 (93.1%) permanently employed in the integrated textile industry (Table 1).

### 3.2. Behavioral and Psychosocial Characteristics of Respondents

Out of the total respondents, five drank alcohol, no one chewed khat, and one smoked cigarettes. The overall use of personal protective equipment (PPE) was low, 57 (19.6%). Out of the total PPE users, the majority (45.6%) used only overalls (“safety tuta”) and covered shoes (Table 2).

### 3.3. Work Environment Characteristics

Respondents were asked about the selected working conditions. Among participants, 61.2% were garment section workers, 74.6% used a machine for their activities, and 95.5% spent less than 48 h in their job per week. More than 70% of workers reported inadequate OSH policies and procedures, awareness, and empowerment (Table 3).

### 3.4. Non-Fatal Occupational Injuries

The prevalence of non-fatal occupational injury was 11% [95 CI: 7.7–15.5%] in the past 12 months. About half (56.3%) of the injuries happened due to contact with machines during normal operations (87.5%). Cuts were the most prevalent type of injury (37.5%) and hand/fingers were the most affected body parts (81.3%) (Table 4).

### 3.5. Factors Associated with Non-Fatal Occupational Injuries

Each variable was analyzed in the bivariate logistic regression analysis to select a candidate for multivariable logistic regression. Socio-demographic variables (sex, age, marital status, and monthly income), behavioral and psychosocial characteristics (sleeping hours, workplace stress, and job satisfaction) and work environment (work section, machine-based work, hazard status, PPE availability, empowerment protection, and obstacle-free floor) were variables with a *p*-value of less than 0.3. From the multiple regression analysis, statistically significant differences in injuries (*p* < 0.05) were observed due to variations in gender, age group, sleeping hours per day, job category, empowerment to prevent accident, and obstacle status of floor (Table 5). Male workers had over three times the chance of sustaining an injury, less than seven hours of sleep in a night had more than two times the likelihood of injury, machines-based workers were more than three times as likely to sustain an injury, and workers who reported inadequate empowerment to prevent accident of injury were four times more likely to sustain a workplace injury compared with their reference group.

## 4. Discussion

The prevalence of non-fatal occupational injuries in this study was 11% in the last 12 months. The current finding shows a lower prevalence than studies in Bangladesh, 28.3% [4], in Ethiopia, 31.4% and 42.7% [9,10], and in Turkey, 65.8% [11]. The reason for this inconsistency might be due to the time period when the studies were conducted. The current study was conducted during the COVID-19 pandemic period. This explains the decrease in the overall incidence of injuries compared with the non-COVID-19 pandemic period [39]. The workers’ work trend during the pandemic might be changed and the industry production operational activities might be compromised [40]. Other explanation might be due to distancing that leads to lower interaction among workers and reduces coworker disturbance in the work area [39,41].

The upper limbs, especially the hands and fingers, were the most affected body parts in this study. Similar studies also reported comparable findings among the textile workers in Ethiopia and Turkey [9,10,11,42]. The majority of the duties in textile industries were carried out with the use of machines. To operate these machines, the body’s upper extremities are used frequently and are more susceptible to injury. The majority of the injuries were one-time occurrences. It is similar to previous research on workplace injuries [9,11]. This could be because the workers learn preventive behavior after an injury so that the injuries might not happen again and again [43].

Gender, age group, sleeping hours, job category, the workplace housekeeping, and empowerment to prevent accidents were significantly associated with non-fatal occupational injuries in the present study.

Male workers had over three times the risk of sustaining an occupational injury [AOR: 3.40; 95% CI (1.13–10.5)] compared to females. This finding was similar to studies among textile industry workers in Ethiopia, any workers in Tanzania and Mexico whereby male workers were found to have more physical injury than females [10,15,44,45]. The reason might be due to male involvement in risk-taking behaviors such as poor application of safety in the work area and engagement with risk work compared to female workers [46,47,48]. Another explanation might be that female workers are usually assigned to less hazardous sections such as knitting and garment jobs rather than weaving and dying jobs [10,42]. The 18–29-year-old age group had greater odds of sustaining an occupational injury [AOR: 6.69; 95% CI (1.35–32.7)] compared to workers 45 years and older. This finding is in line with a similar study among textile workers [9]. This is most likely due to the higher risk jobs that younger workers carry out and also because they have less experience on how to prevent injury compared to older workers [49].

According to different studies in Ethiopia, Korea, and California, drinking alcohol, khat chewing, and cigarette smoking were risk factors that increase the chance of injuries at work [20,50,51]. The current study’s findings revealed that few respondents reported about these practices. The explanation for low use of alcohol, khat, and cigarettes might be due to the code of ethics in the industry. Substance use such as these are restricted in this industry. It is also one of the recruiting criteria for workers. Seven or more hours of sleep per night is recommended to promote optimal health among adults aged 18 to 60 years [52,53]. According to the current study, workers who reported sleeping less than seven hours per night had more than two times the likelihood of sustaining an injury [AOR: 2.67; 95% CI (1.03–6.97)] compared with than who reported at least seven or more hours of sleep each night. This finding aligns with other work-related injury studies in the United States of America and Japan [54,55]. Lack of sleep, poor quality sleep, sleep disorders, and sleep disturbances increase the likelihood of sustaining a work-related injury [56,57].

Workers who use machines to manufacture textiles products were more than three times likely [AOR: 3.59; 95% CI (1.02–12.6)] to sustain a workplace injury compared with non-machine-based jobs. Contacting machinery without appropriate safety measures may be the reason for this finding [5,58,59]. This study also showed that poor workplace housekeeping was associated with injury [AOR: 5.87; 95% CI (1.45–23.8)]. Working in the unsafe work area due to poor housekeeping might lead to an accident while carrying out a usual job [59].

OSH empowerment refers to the extent to which individuals feel free to voice their health and safety concerns, question health and safety, and refuse hazardous work [22]. Worker empowerment or authority to participate in injury and illness prevention is important to reduce injury in any industry. Textile industry workers in this study who reported inadequate empowerment to prevent accidents had an injury risk of four times greater [AOR: 4.6; 95% CI (1.01–20.9)] than reported adequate empowerment protection. Textile workers may be unaware of their employment rights and might not be able to raise safety concerns with their bosses [14,60]. A study indicated that empowerment protection such as the individual’s ability to feel free to voice health and safety concerns, ask questions about health and safety, and refuse unsafe duties was related to lower injury rates [22].

The strengths of this study are good participation rate, the use of standard measurement tools, and collecting through face-to face interviews. However, limitations of this study include that the findings were based on self-reports and injuries were not clinically confirmed. This might lead to recall bias and may underestimate the injury prevalence. Finally, the referred sample values that presented injuries, although the statistical parameters were significant, are only representative in the study and cannot be extrapolated to other work contexts.

## 5. Conclusions

The findings in this study have provided an insight into the injuries that typically occur among textile industry workers. Generally, the prevalence of non-fatal occupational injuries among workers is lower than the previous studies among textile workers. The upper limbs were the most affected body regions and gender, age, sleeping hours per night, job category, the workplace housekeeping, and empowerment to prevent accidents were significantly associated with injury. Therefore, workplace safety improvements, behavioral change in sleeping patterns, and worker empowerment to participate in injury prevention should be focus areas in injury intervention.

## Figures and Tables

**Table 1 ijerph-19-03688-t001:** Distribution of socio-demographic among integrated textile industry workers, Addis Ababa Ethiopia, 2021.

Variables Categories	Frequency (*n* = 291)	Percentage
**Sex**		
Male	77	26.5
Female	214	73.5
**Age**		
<30	171	58.8
≥30	120	41.2
**Marital status**		
Married	115	39.5
Single/widow/divorced	176	60.5
**Educational Status**		
less than Grade 8	99	34.0
Secondary school (9–12)	128	44.0
Graduated from vocational school and above	64	22.0
**Service year**		
<3 Years	113	38.80
3–6 Years	84	28.90
>6 Years	94	32.30
**Employment status**		
Permanent worker	271	93.1
Temporary/Contract worker	20	6.9
**Monthly net income in US Dollar**		
≤35.5	109	37.5
35.51–75.0	152	52.2
>75.0	30	10.3

**Table 2 ijerph-19-03688-t002:** Behavioral and psychosocial characteristics of respondents in the integrated textile industry, Addis Ababa Ethiopia, 2021.

Variables Categories	Frequency (*n* = 291)	Percentage
**Personal Protective Equipment (PPE) use**		
Yes	57	19.6
No	234	80.4
**Reasons for non-use of PPE (*n* = 234)**		
Unavailability	184	78.6
Lack of training	20	8.5
Discomfort	14	6.0
Not required, not fitted to the body, limits work performance, etc.	50	21.4
**Self-reported job satisfaction**		
Yes	223	76.6
No	68	23.4
**Work-related stress**		
Yes	27	9.3
No	264	90.7
**Sleeping hours in a night**		
<7	65	22.3
≥7	226	77.7

**Table 3 ijerph-19-03688-t003:** Work environment characteristics of the integrated textile industry workers, Addis Ababa, Ethiopia, 2021.

Variables Categories	Frequency (*n* = 291)	Percent (%)
**Workplace**		
Knitting	18	6.2
Dying	22	7.6
Garment	178	61.2
Blanket	66	22.7
Maintenance and mechanic	7	2.4
**Working hours per week**		
≤48 h	278	95.5
>48 h	13	4.5
**Adequate workspace**		
Yes	274	94.2
No	17	5.8
**Adequate light**		
Yes	286	98.3
No	5	1.7
**Obstacle-free floor**		
Yes	267	91.8
No	24	8.2
**Machine-based job**		
Yes	217	74.6
No	74	25.4
**Guarded machine**		
Yes	206	70.8
No	85	29.2
**Exposed to at least two hazards**		
Yes	100	34.4
No	191	65.6
**Occupational health and safety (** **OSH** **)policies and procedures** **protection**		
Adequate	25	8.6
Inadequate	266	91.4
**OSH awareness** **protection**		
Adequate	85	29.2
Inadequate	206	70.8
**OSH empowerment** **protection**		
Adequate	60	20.6
Inadequate	231	79.4

**Table 4 ijerph-19-03688-t004:** Characteristics of injuries among integrated textile industry workers, Addis Ababa, Ethiopia, 2021.

Variables	Frequency	Percent (%)
**None-fatal occupational injury (*n* = 291)**		
Yes	32	11
No	259	89
**Occupational injury rate among section (*n* = 291)**		
Knitting (*n* = 18)	1	5.6
Dying (*n* = 22)	5	22.7
Garment (*n* = 178)	14	7.9
Blanket (*n* = 70)	11	16.7
Utility (*n* = 7)	1	14.3
**Injury frequency (*n* = 32)**		
One time	30	10.3
Two and more	2	0.7
**Parts of the body affected (*n* = 32)**		
Fingers	15	46.9
Hand	11	34.4
Foot	3	9.4
Eye, Back, Knee	3	9.4
**Cause of injury (*n* = 32)**		
Machine-based activities (operating of any machine)	28	87.5
Non-machine-based activities (cleaning, sorting collecting product and maintenance)	4	12.5

**Table 5 ijerph-19-03688-t005:** Multivariate analysis of individual factors associated with non-fatal occupational injury among integrated textile industry workers, Addis Ababa Ethiopia, 2021.

Variable Name	Injury Status (*n* = 291)		Crude OR (95%CI)	Adjusted OR (95%CI)
	Yes (%)	No (%)		
**Sex**				
Male	16 (20.8)	61 (79.2)	3.23 (1.53–6.87) *	3.4 (1.13–10.5) *
Female	16 (7.5)	198 (92.5)	1	1
**Age category**				
18–29	20 (11.7)	151 (88.3)	2.43 (0.69–8.49)	6.69 (1.35–32.7) *
30–45	9 (14.5)	53 (85.5)	0.78 (0.33–1.82)	0.98 (0.34–2.84)
>45	3 (5.2)	55 (94.8)	1	1
**Marital Status**			1	1
Married	17 (14.8)	98 (85.20)	1.86 (0.89–3.90)	2.43 (0.94–6.27)
Single/widowed/divorced	15 (8.5)	161 (91.50)	1	1
**Sleeping hour in a day**				
<7	11 (16.9)	54 (83.1)	1.99 (0.90–4.38)	2.67 (1.03–6.97) *
≥7	21 (9.3)	205 (90.7)	1	1
**Workplace stress status**				
Yes	54 (48.2)	58 (51.8)	2.00 (0.70–5.70)	0.72 (0.18–2.90)
No	44 (40.0)	66 (60.0)	1	1
**Job satisfaction status**				
Yes	21 (9.4)	202 (90.6)	1	
No	11 (16.2)	57 (83.8)	1.86 (0.85–4.08)	0.55 (0.21–1.42)
**Work section**				
Knitting	1 (5.6)	17 (94.4)	1	1
Dying	5 (22.7)	17 (77.3)	2.83 (0.15–52.7)	0.94 (0.03–33.1)
Garment	14 (7.9)	164 (92.1)	0.57 (0.56–5.88)	0.34 (0.22–5.12)
Blanket	11 (16.7)	55 (83.3)	1.95 (0.22–17.38)	0.38 (0.021–7.13)
Utility	1 (14.3)	6 (85.7)	0.83 (0.09–7.63)	0.39 (0.03–5.87)
**PPE availability**				
Yes	12 (14.5)	71 (85.5)	1.59 (0.74–3.42)	1.99 (0.72–5.44)
No	20 (9.6)	188 (90.4)	1	1
**Machine-based Job**				
Yes	28 (12.9)	189 (87.1)	2.60 (0.88–7.66)	3.99 (1.05–15.2) *
No	4 (5.4)	70 (94.6)	1	
**Obstacle-free floor**				
Yes	25 (9.4)	242 (90.6)	1	
No	7 (29.2)	17 (70.8)	3.99 (1.51–10.5) *	5.87 (1.45–23.8) *
**Hazard exposure status**				
Yes	26 (13.6)	165 (86.4)	1	
No	6 (6.0)	94 (94.0)	2.47 (0.98–6.21)	0.44 (0.16–1.19)
**Empowerment protection**				
Adequate	3 (5.0)	57 (95.0)	1	
Inadequate	29 (12.6)	202 (87.4)	2.73 (0.80–9.28)	4.6 (1.01–20.9) *

COR, crude odds ratio; AOR, adjusted odds ratio; CI, confidence interval. * Statistically significant difference (*p* < 0.05).

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
