# Peer review of "Non-Fatal Occupational Injury Prevalence and Associated Factors in an Integrated Large-Scale Textile Industry in Addis Ababa, Ethiopia"

_ijerph, 2022, doi:10.3390/ijerph19063688_

Round 1

Reviewer 1 Report

Interesting research and good presentation of the document

Strength

The document has a good structure and content.

Weakness

It is suggested to broaden the discussion with previous studies, as well as its practical implications with the sector studied and/or with the employees of this type of company. It is also important to extend the conclusion with the highlights of the study.

Author Response

We would like to thank the reviewers for the time and effort put on to revise our manuscript in detail. We believe that the comments have identified important areas which required improvement. After completion of the suggested edits, the revised manuscript has benefitted from an improvement in the overall presentation and clarity. We tried to address the issue based on your comment.

Reviewer 2 Report

I would like to congratulate the authors for dimensioning a problem that is so widespread and little addressed in countries with high marginalization, these aspects are rarely seen in other parts of the world and often only focus on high-income countries.

Below I share my observations regarding the manuscript.

Lines 30 and 31, briefly indicate what type of injuries they are, as well as their prevalence.

In section 2.2 Define if the sample size was calculated, as well as the type of sampling used.

In section 2.4 Data collection tool and procedure, please define how much time was allocated to answer the applied instruments?

In lines 73 and 74 authors give very general information, please specify which questionnaire or questionnaires they refer to.

Line 75-77 were the questionnaires validated for the study population? Define statistical validation values ​​(i.e. Cronbach's alpha), if not, how do they justify their use?

Line 77 Briefly define the dimensions and cut-off points of this instrument.

In line 93 where authors indicate the collection of data, define if those areas corresponded to their workplaces or the placer where this activity was realized.

Line 99. What was the justification for not using informed consent?

Line 117. Job satisfaction status, what dimensions does it measure, how many questions have the instrument?

Line 119. khat chewing. Clarify for the unfamiliar audience why this item is important?

Line 136, “A total of 291 (93.6%), 26.5% male and 73.5% female participants were interviewed.” Number 291 it is confusing since in the study population section authors mention that there were 311 participants, if there were 311 participants clarify why only 291 answered.

Table 1. Monthly net income Ethiopian birr. It would be very helpful to make a conversion to dollars so that the audience not familiar with the Ethiopian currency could measure this amount more directly.

It is noteworthy that of the total number of respondents, five drank alcohol, no one chewed khat and one smoked cigarette. How do the authors explain the low prevalence of these activities? It would be important to point it out in the discussion.

Line 174 I suggest clarifying the reason for the low incidence of injuries during the COVID-19 pandemic.

Line 189 please clarify what risk behaviors in men refer to.

In the discussion it would be convenient to discuss the role of income since although the authors apparently did not find an association in it, is this always the case? or what is the impact of income on the presence of accidents.

The authors need to delve deeper into the limitations of their study, one of them is the referred sample value that presented injuries, although the statistical parameters were significant, these are only representative in the study and cannot be extrapolated to other work contexts, the instruments were validated for the study population? Was the time to answer enough to obtain reliable data?

Author Response

We would like to thank the reviewers for the time and effort put on to revise our manuscript in detail. We believe that the comments have identified important areas which required improvement. After completion of the suggested edits, the revised manuscript has benefitted from an improvement in the overall presentation and clarity.

Reviewer 3 Report

Please see attached

Author Response

(The authors gave the same response as above.)

Reviewer 4 Report

Dear Authors, 

I really Appreciate your work in the safety of Textile industry and injury factors. Can you address the comments attached for better value and presentation. 

Regards, 

Author Response

(The authors gave the same response as above.)
